# Assessing the role of the pelvic canal in supporting the gut in humans

**Jeanelle Uy**[1]*, **Natalie M. Laudicina**[2]

**1** Anthropology, California State University Long Beach, Long Beach, California, United States of America,
**2** Biomedical Sciences, Grand Valley State University, Allendale, Michigan, United States of America

* jeanelle.uy@csulb.edu

## Abstract

The human pelvic canal (true pelvis) functions to support the abdominopelvic organs and serves as a passageway for reproduction (females). Previous research suggests that these two functions work against each other with the expectation that the supportive role results in a narrower pelvic midplane, while fetal passage necessitates a larger opening. In this research, we examine how gut size relates to the size and shape of the true pelvis, which may have implications on how gut size can influence pelvic floor integrity. Pelves and *in vivo* gut volumes were measured from CT scans of 92 adults (48 female, 44 male). The true pelvis was measured at three obstetrical planes (inlet, midplane, outlet) using 11 3D landmarks. CT volumetry was used to obtain an individual's gut size. Gut volume was compared to the pelvic planes using multiple regression to evaluate the relationship between gut size and the true pelvis. We find that, in males, larger gut sizes are associated with increased mediolateral canal dimensions at the inlet and midplane. In females, we find that larger gut sizes are associated with more medially-projecting ischial spines and an anteroposteriorly longer outlet. We hypothesize that the association of larger guts with increased canal width in males and increased outlet length in females are adaptations to create adequate space for the gut, while more medially projecting ischial spines reduce the risk of pelvic floor disorders in females, despite its possible spatial consequences for fetal passage.

## Introduction

The morphology of the human pelvic canal is a compromise between having enough space for parturition (females) and the abdominopelvic viscera, while also having substantial bony support for the pelvic floor, which supports the abdominopelvic weight. Additionally, locomotor constraints may influence canal size directly or indirectly through changes in other parts of the pelvis [1]. These roles of the pelvis can come into conflict with each other, as increases in the pelvic canal size can compromise the integrity of the pelvic floor [2–4]. The pelvic floor hypothesis [5] states that human ischial spines project medially because they serve as anchor points for the pelvic diaphragm. The ischial spines project medially, regardless of sex, because the pelvic floor contracts to support the abdominopelvic organs and reduce the risk of pelvic floor disorders (PFDs). Thus, the pelvic canal serves multiple roles in terms of visceral support:

**Competing interests:** The authors have declared
that no competing interests exist.

a space that must accommodate the abdominopelvic organs, a bony support for abdominopelvic weight, and a bony anchor for the pelvic floor.

The pelvic girdle can be divided into two components: the false and true pelvis. The false pelvis is the superior portion of the girdle including the iliac blades. One function of the iliac blades is to accommodate and support viscera. In hominin evolution, the pelvis adapted to bipedal gait by repositioning the iliac blades more sagittally and laterally [1]. This morphological change also accommodated the increased viscera volume hypothesized in early hominins [1]. The true pelvis consists of the canal portion surrounded by bony supports and helps the false pelvis support abdominopelvic viscera, while in females also provides a passageway for reproduction [1]. Measures of the canal (Fig 1: inlet, midplane, and outlet) suggest that in terms of pelvic floor integrity, the size and shape of the anteroposteriorly (AP) constricted inlet and the mediolaterally (ML) constricted midplane may have the greatest influence on helping to support the abdominopelvic viscera.

The abdomen is spatially dominated by the volume of the small and large intestines ("gut volume" or "gut size"); therefore, its size contributes to much of the abdominopelvic weight. Biological anthropologists speculate that the morphology of the pelvis can reflect the size of the gastrointestinal tract; this hypothesis has partially been the basis for theories about the evolution of diet, locomotion, and brain size [6–8]. Uy, Hawks, and VanSickle [9] examined the association between gut volume and the whole pelvis in humans and found correlations between bispinous distance and outlet width and gut size in males, but not females. Additionally, they found that gut volume has no relationship with body weight in females, but scales with slight negative allometry in males. There seems to be a dynamic relationship between the gut (and perhaps other abdominal organs) and the pelvis that may be confounded by sex differences.

An additional aspect of pelvic morphology is the ability of the pelvis to adapt to a variety of factors such as birth, locomotion, and climate. This plasticity is in part due to the relative delay in complete ossification of the pelvic girdle components (two ossa coxae and one sacrum). Each os coxa is composed of three bones: the ilium, ischium, and pubis. These pelvic components begin to develop early in utero, during the fifth week of gestation [10]. Ossification occurs every month, continuing throughout childhood and adolescence, not obtaining complete fusion until adulthood [10]. Hormones can also influence the pelvic canal shape, especially in reference to obstetric dimensions in females [11–16]. Throughout a woman's lifetime, fluctuations in estrogen levels can result in changes in the pelvic canal's morphology, and the sex-specific morphologies may be in part due to the increased estrogen levels seen in female neonates, starting in utero [12, 13, 15].

Due to its prolonged ossification period, the pelvis can be subjected to many effects, influencing its morphological structure. In this research, we examine one such influence: how gut size correlates with the size and shape of the true pelvis. Changes in pelvic morphology due to gut size may then have implications on how abdominal weight can influence pelvic floor integrity.

In this study, we test Abitbol's [5] pelvic floor hypothesis that states that pelvic canal morphology (e.g., ischial spines) is for visceral support. To do so, we evaluate how size and shape of the inlet, midplane, and outlet are influenced by increased gut volume. We hypothesize that a larger gut volume will result in decreased pelvic canal dimensions to allow for more structural support of the increased mass. We extend Abitbol's [5] original pelvic floor hypothesis to include an examination of how sex differences in the pelvic canal may allow for different visceral support mechanisms. Tague and Lovejoy [1] state that in males, the true pelvis is adapted for visceral support, while the female pelvis is adapted for both parturition and visceral support. Female pelves, it is hypothesized, must remain as constricted as possible to reduce strain

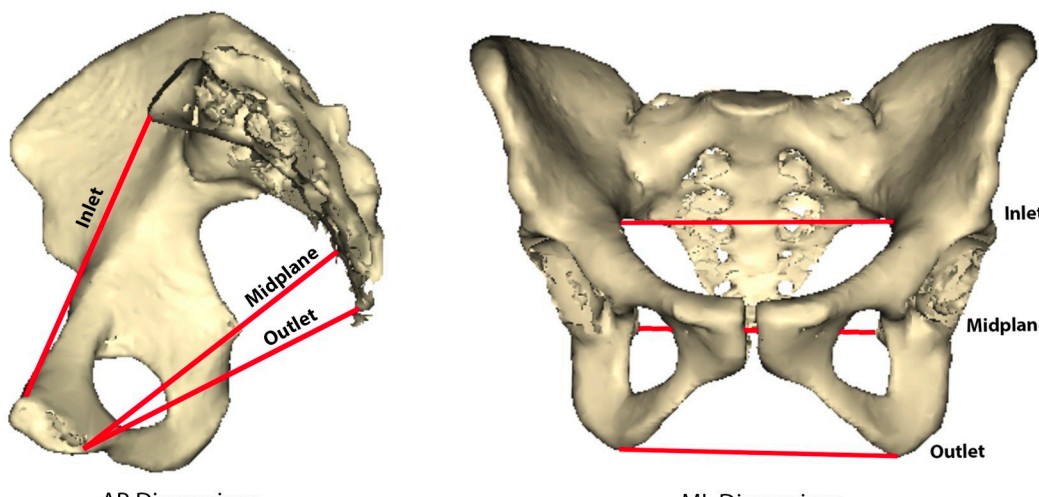

**Fig 1. Two views of the pelvis.** One showing the anteroposterior (AP) dimensions of the inlet, midplane, and outlet, and the other showing the mediolateral (ML) dimensions of the inlet, midplane, and outlet.

on pelvic floor muscles, while also retaining enough space for fetal passage [2–4, 17, 18]. This research contributes to the understanding of the evolution and function of pelvic canal morphology.

## Materials and methods

### Study sample

We analyzed a total of $N = 92$ adults, 44 males and 48 females. The abdominal computed tomography (CT) scans were performed for routine purposes in Madison, Wisconsin and the subjects included had no observed intestinal pathologies or signs of previous surgical alterations. The sample consists of individuals aged 18–25; as a narrow age range may control for variation in gut and pelvic size related to age [15, 19, 20]. To increase visibility of the intestines, the subjects were given an oral contrast prior to the scan. The abdominal scans of living adults used in this study are secondary data. The scans were originally obtained for another study done by one author (JU) where they were collected from an archival database at the University of Wisconsin-Madison School of Medicine and Public Health. As the scans were already previously de-identified, the IRB did not require a new review for this present study. A summary of the sample's body weight, stature, and gut volume can be found in Table 1.

### Data collection

In order to analyze the canal's shape and dimensions, we extracted 3D models of the pelvis from the CT scans using *Horos* (www.horosproject.com). From these models, we obtained 3D landmarks on the pelvic canal using *Fiji* [21]. Landmark definitions can be found on Fig 2 and

**Table 1. Descriptive statistics for study sample.**

|  | Males (n = 44) | Females (n = 48) |
|---|---|---|
| **Body Weight (kg)** | 80.43 ± 19.27 | 67.17 ± 13.15 |
| **Stature (m)** | 1.77 ± 0.07 | 1.64 ± 0.07 |
| **Gut Volume (cc)** | 5384 ± 1320.86 | 4192.13 ± 912.44 |

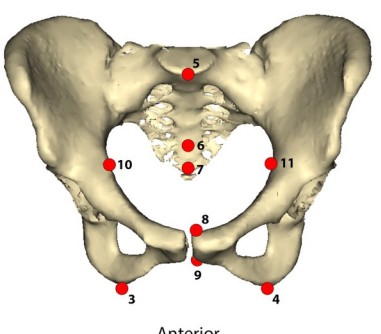 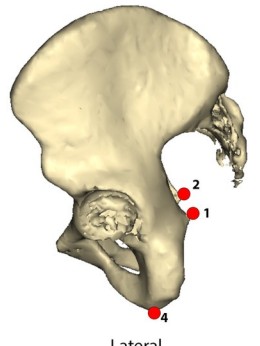 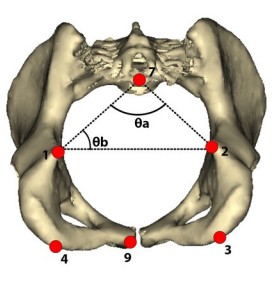

Anterior Lateral Inferior

**Fig 2. Canal landmarks.** Eleven landmarks, shown as red circles, were taken on 3D models of pelves to represent the canal. One pelvis is shown here from the anterior, lateral, and inferior points of view.

Table 2. To obtain linear anteroposterior (AP) and mediolateral (ML) dimensions of the inlet, midplane, and outlet, interlandmark distances (ILDs) were calculated using *Geomorph* for R [22]. These dimensions are described in Table 3. The 2D shape of the inlet, midplane, and outlet was quantified by dividing the anteroposterior dimension by the mediolateral dimension of each plane.

Observer error in ILD and gut volume (GV) measures was calculated using the intraclass correlation coefficient (ICC) [24], which has been used to assess error in both bony and soft tissue digital measurements [25]. ICC estimates were calculated using the *psych* package in R [26] based on absolute agreement, two-way mixed effects model with a 95% tolerance level. This model shows the strength of inter- and intraobserver agreement on a measurement from one specimen. Most ICCs showed good ($\geq$0.75) to excellent ($\geq$0.90) agreement on ILDs between the observers and within each observer. The exception was outlet ML diameter (OML); consequently, the landmarks used to obtain OML were classified as "fuzzy" landmarks [23, 27]. OML was measured twice; the average values were used for analysis. GVs were measured by only one author (JU) and they showed excellent agreement (ICC = 0.94).

Abitbol [5] suggests that a narrower midplane is better suited to support visceral weight. In order to quantify the medial projection of the ischial spines, we calculated the angle between the lines formed from the apex of the fifth sacral vertebra to the left and right ischial spines (Fig 2, inferior view). To do this, we measured the interlandmark distances (in R) between the apex of the fifth sacral vertebra to the left ischial spine and to the right ischial spine, and the

**Table 2. Landmark descriptions.**

| No. | Landmark | Description |
|---|---|---|
| 1, 2 | Ischial spine | Dorsal-most projection of the spine located on the posterior ischium |
| 3, 4 | Ischial tuberosity ^ | Left and right inferior-most point of the ischial tuberosity |
| 5 | Sacral promontory | Midpoint of sacral promontory |
| 6 | Between S4 and S5 | Midpoint between 4th and 5th sacral vertebrae |
| 7 | Apex of S5 | Apex of 5th sacral vertebra |
| 8 | Dorsal superior pubic symphysis | Most superior medial point of the dorsal pubis, left |
| 9 | Dorsal inferior pubic symphysis | Most inferior medial point of the dorsal pubis, left |
| 10, 11 | Max. distance between iliopectineal lines | Widest transverse dimension of the inlet |

^ indicates "fuzzy" landmark [23]

**Table 3. Canal measurements.**

| Canal Dimension | Landmarks | Description |
|---|---|---|
| Inlet AP | 5–8 | Anteroposterior dimension of inlet: distance between promontory and dorsal superior pubic symphysis |
| Inlet ML | 10–11 | Mediolateral dimension of inlet: maximum distance between iliopectineal lines |
| Midplane AP | 6–9 | Anteroposterior dimension of midplane: distance from between S4 and S5 to the dorsal inferior pubic symphysis |
| Midplane ML | 1–2 | Mediolateral dimension of the midplane: distance between the left and right ischial spines |
| Outlet AP | 7–9 | Anteroposterior dimension of the outlet: distance between the apex of S5 to dorsal inferior pubic symphysis |
| Outlet ML | 3–4 | Mediolateral dimension of the outlet: distance between ischial tuberosities |
| Medial projection of ischial spines | $\theta_a$ | Angle between the straight lines formed by the left and right ischial spines to the apex of S5 |
| S5 Position | $\theta_b$ | Angle between the straight line formed by the left ischial spine to the apex of S5 and the straight line formed by the left ischial spine to the right ischial spine |

Canal measurements were obtained from 3D models based on landmarks shown in Fig 2. Full landmark descriptions are in Table 2.

distance between the left and right ischial spines. From this "triangle", we were able to calculate the angle ($\theta_a$) that reflects the degree of medial projection of the ischial spines. A smaller angle represents more medially projecting spines (a narrower midplane) while a larger angle represents less medially projecting spines (a wider midplane). We specifically chose to measure this angle in addition to calculating the distance between the ischial spines (MML) because the MML only captures the absolute distance between ischial spines, but not how much they project into the pelvic canal. That is, two people could have the same MML but different $\theta_a$. Abitbol [5] directly comments on ischial spine projection as an important component of pelvic floor integrity.

To further examine the size of the pelvic floor area, we also calculated the angle that reflects the projection of the S5 apex into the pelvic canal ($\theta_b$ in Fig 2). Using the same "triangle" described in the previous paragraph, we determined the angle between the linear distance from the S5 apex and the left ischial spine and the linear distance between the left and right ischial spines. This angle ($\theta_b$) measures how much the sacrum projects towards the ischial spines. A smaller angle indicates that the S5 projects more prominently into the pelvic canal. A more posterior position (a larger $\theta_b$) of the S5 may indicate larger pelvic floor area. We expect that this angle will differ between sexes because the posterior space of the pelvic canal has been shown to be sexually dimorphic [28, 29].

To obtain gut size, we used CT volumetry, a technique commonly used in radiology that has been described since the 1970's [30], to obtain intestinal volume using *Fiji* [21]. Intestinal volume, or GV, is the volume of both the small and large intestines. On each image slice from the CT scan, the area of the visible intestines was measured and then multiplied by the thickness of that slice (5 mm). This calculation gives the volume of the intestines for that slice. The same calculation is repeated for all slices that contain the intestines. The sum of these volumes is the GV for an individual.

## Statistical analysis

Multiple linear regression analyses were conducted to assess the relationship of the canal and gut size, with body weight as a covariate. Statistical analyses were conducted using R [31]. All

values were log-transformed prior to analyses. Previous work has established that body weight has no relationship with GV in females, but scales with negative allometry in males [9]. Therefore, we used body weight as an interacting predictor variable in the male sample (canal~GV*-weight) and as a non-interacting predictor variable in the female sample (canal~GV+weight). In other words, in males, the regression model accounts for body weight's impact on GV when assessing both variables' relationship with the pelvic canal measurement. In females, the regression model treats GV and body weight as independent. Additionally, using weight as an interacting covariate (instead of non-interacting) does not have an effect on the female regression model when tested with an ANOVA. When assessing the significance of the results, we adopted a conservative alpha of 0.01, though we also note in Table 4 if $p \leq 0.05$.

In order to disentangle the relative importance of body mass and gut volume on pelvic dimensions, we performed a *post hoc* analysis using the method of relative weights [32, 33]. The R function *relweights* [33] determines the approximate percentage of $R^2$ that is accounted for by each predictor variable (GV or body weight).

## Results

Our findings show that overall there are some notable differences between the sexes in the relationship of gut volume and the true pelvis. In order to better understand these relationships, we analyzed each canal dimension and shape by sex. Table 4 summarizes the results of the regressions and Table 5 summarizes the relative importance of each predictor variable in accounting for the $R^2$ value.

### Inlet

The shape of the inlet in the male sample exhibits a significant relationship with GV and body mass. The regression model shows that GV and body mass accounts for about 25% of the variation in inlet shape ($R^2 = 0.25$). The relative importance of GV (61.5%) in accounting for that $R^2$ is higher than body mass (38.5%). Fig 3 illustrates how the larger-bodied males tend to have a more pronounced change in inlet shape as gut volume increases. In particular, it seems that the inlet becomes relatively mediolaterally wider as gut size increases, and this effect is more pronounced in larger-bodied males and diminished in smaller-bodied males.

Notably, GV and body mass have no detectable effect at all with inlet shape in females (Fig 3). Additionally, neither male nor female samples exhibit significant associations between the linear dimensions of the inlet and GV and body mass.

### Midplane

The effect of GV and body mass on midplane shape is low in both males and females, explaining about 19% and 17% of variation, respectively. Both the male and female samples have significant relationships between midplane shape and GV and body mass, but the pattern of relationships is different.

In the male sample, GV accounts for 77.2% of $R^2$. Larger-bodied males have a midplane shape that becomes transversely wider as GV increases, but this effect is not as pronounced in smaller-bodied males (Fig 4). In the female sample, GV accounts for only 4.8% of $R^2$. Body size seems to have a larger effect on midplane shape within females; there is very little change in midplane shape related to GV change (Fig 4). However, it seems that female individuals with larger GVs tend to have slightly wider midplanes. Neither male nor female samples exhibit significant associations between midplane linear dimensions and GV and body mass. However, MAP in females and MML in males do show significant $R^2$ values if using the less conservative $\alpha = 0.05$.

**Table 4. Summary of regression results.**

| Variables | $\beta_{GV}$ (p-value) | $\beta_{BodyMass}$ (p-value) | $R^2$ | Regression Model p-value |
|---|---|---|---|---|
| **Males** | | | | |
| IAP | 1.43 (0.14) | 2.86 (0.13) | -0.01 | 0.47 |
| IML | -0.61 (0.44) | -1.34 (0.39) | 0.06 | 0.16 |
| Inlet (AP/ML) | 2.66 (**0.002**\*\*) | 5.39 (**0.002**\*\*) | 0.25 | **0.002**\*\* |
| MAP | 0.08 (0.93) | 0.23 (0.86) | -0.05 | 0.80 |
| MML | -1.51 (0.08) | -3.15 (0.06) | 0.16 | **0.02**\* |
| Midplane (AP/ML) | 1.95 (**0.05**\*) | 4.15 (**0.04**\*) | 0.19 | **0.01**\*\* |
| OAP | -0.32 (0.74) | -0.44 (0.81) | -0.03 | 0.65 |
| OML | -0.35 (0.80) | -1.13 (0.68) | 0.05 | 0.17 |
| Outlet (AP/ML) | 0.80 (0.57) | 2.18 (0.45) | 0.07 | 0.12 |
| Ischial Spine Projection | 0.33 (0.76) | 0.36 (0.86) | -0.03 | 0.59 |
| S5 Position | -0.35 (0.85) | -0.30 (0.94) | -0.04 | 0.75 |
| **Females** | | | | |
| Variables | $\beta_{GV}$ (p-value) | $\beta_{BodyMass}$ (p-value) | $R^2$ | Regression Model p-value |
| IAP | 0.029 (0.63) | 0.032 (0.65) | -0.03 | 0.69 |
| IML | 0.074 (0.11) | 0.030 (0.57) | 0.05 | 0.13 |
| Inlet (AP/ML) | -0.044 (0.47) | 0.0020 (0.98) | -0.03 | 0.74 |
| MAP | 0.038 (0.44) | 0.13 (**0.03**\*) | 0.13 | **0.02**\* |
| MML | 0.059 (0.32) | -0.075 (0.27) | -0.01 | 0.45 |
| Midplane (AP/ML) | -0.021 (0.07) | 0.20 (**0.002**\*\*) | 0.17 | **0.005**\*\* |
| OAP | 0.071 (0.19) | 0.16 (**0.01**\*) | 0.19 | **0.003**\* |
| OML | 0.024 (0.71) | -0.11 (0.15) | 0.004 | 0.34 |
| Outlet (AP/ML) | 0.047 (0.49) | 0.27 (**0.001**\*\*) | 0.24 | **0.0007**\*\* |
| Ischial Spine Projection | -0.11 (0.07) | -0.099 (0.13) | 0.14 | **0.01**\*\* |
| S5 Position | 0.26 (**0.03**\*) | 0.16 (0.23) | 0.16 | **0.008**\*\* |

Results of linear regression analyses comparing pelvic canal measurements and gut volume with body mass as a covariate. The beta coefficients of each predictor variable and their p-values, the $R^2$ values of the models, and p-values of the models are reported. **Bolded** values indicate significance, with \* for $p \leq 0.05$ and \*\* for $p \leq 0.01$.

The degree of medial projection of the ischial spines has a significant relationship with GV and body mass in females, but not in males. The regression model shows that GV and body mass explain about 14% of the variation in female ischial spine projection. In this case, GV

**Table 5. Relative importance of predictor variables on canal shape and dimensions.**

| Significant Predictor Variables | % of $R^2$ accounted for by GV | % of $R^2$ accounted for by Body Weight |
|---|---|---|
| **Males** | | |
| Inlet | 61.5% | 38.5% |
| Midplane | 77.2% | 22.8% |
| **Females** | | |
| Midplane | 4.8% | 95.2% |
| OAP | 31.1% | 68.9% |
| Outlet | 16.1% | 83.9% |
| Ischial Spine Projection ($\theta_a$) | 56.3% | 43.7% |
| S5 Position ($\theta_b$) | 68.1% | 31.9% |

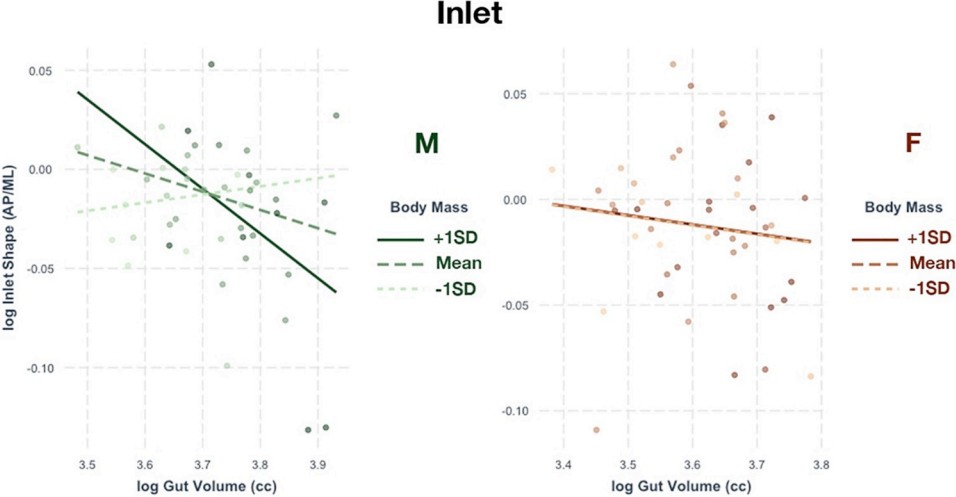

**Fig 3. Relationship of inlet shape with gut volume.** Regressions between gut volume and pelvic canal shape at the inlet relative to body mass. The male sample (left, green) shows a significant relationship. To visualize the effect of body mass on the variables, the plots shown here separate the data points into three groups: those with body masses below 1 SD from the mean (lightest color), those with body masses above 1 SD from the mean (darkest color), and those with body masses between -1 SD and +1 SD from the mean (medium color). The regression lines are the best fit lines for each group.

seems to account for 56.3% of $R^2$ while body weight accounts for 43.7%. As GV increases, the ischial spine projects more medially regardless of body size in females (Fig 5A).

## Outlet

The OAP dimensions and outlet shape both show significant associations with GV and body mass in females, but not in males. GV and body mass explain 19% of the variation in OAP diameter and 24% of outlet shape. As female GV increases, OAP diameter increases and outlet shape expands anteroposteriorly and narrows mediolaterally (Fig 6). However, body mass may

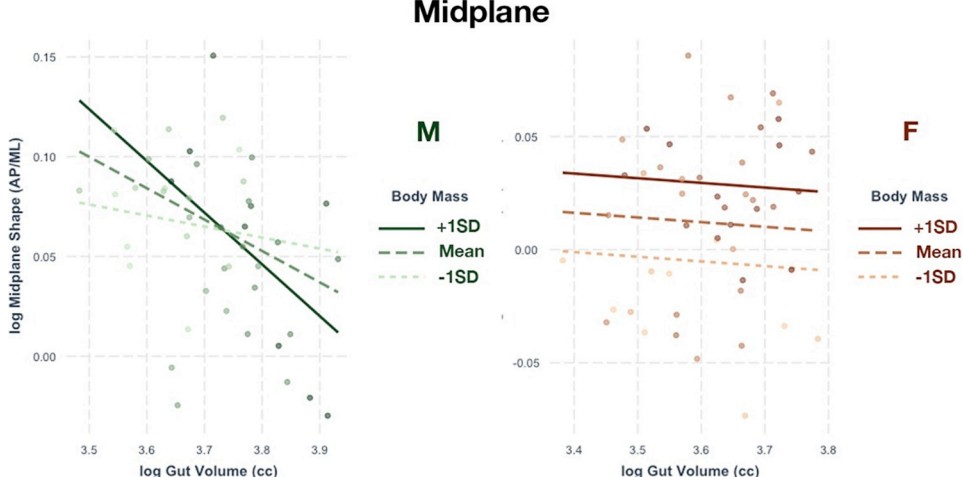

**Fig 4. Relationship of midplane shape with gut volume.** Regressions between gut volume and pelvic canal shape at the midplane relative to body mass. Both the male and female samples (right, orange) show significance.

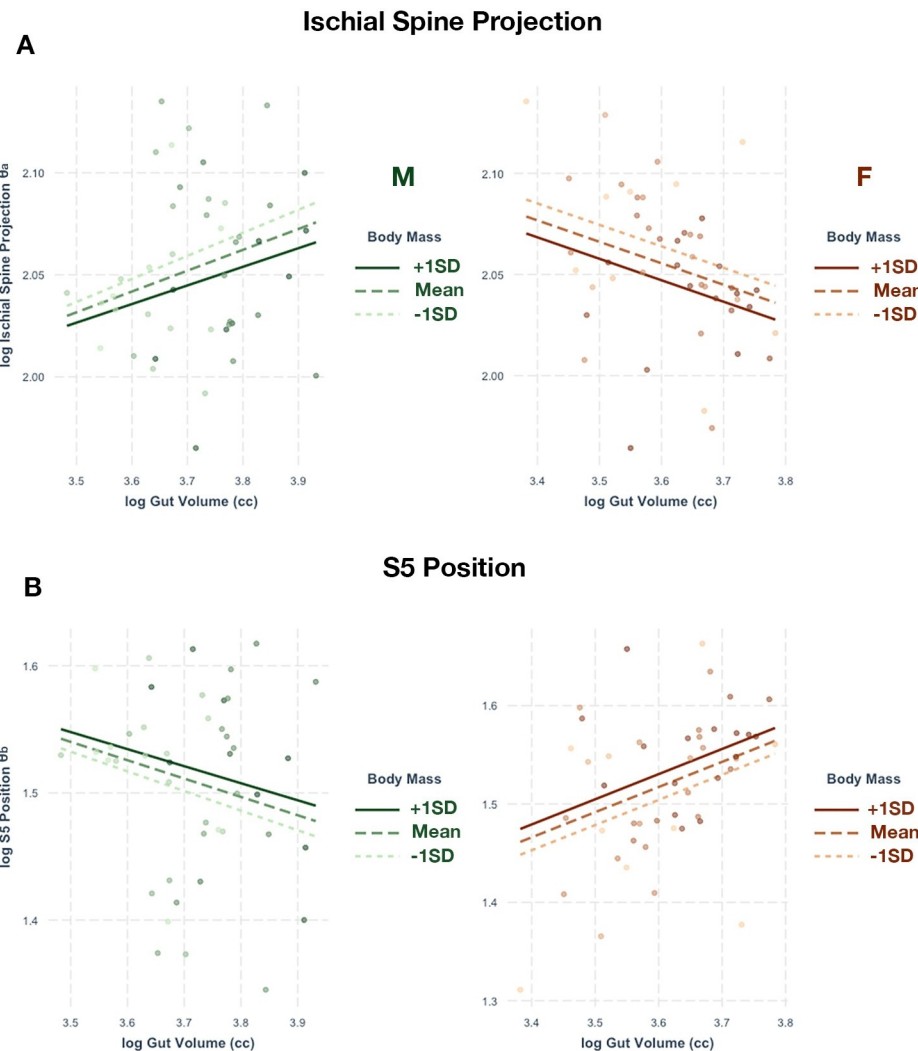

**Fig 5. Relationship of ischial spine and S5 position with gut volume.** Regressions between gut volume and (A) ischial spine projection ($\theta_a$) and (B) S5 apex position ($\theta_b$) relative to body mass. A lower $\theta_a$ indicates more medially-projecting ischial spines. A higher $\theta_b$ indicates a longer distance between the ischial spines and the S5 apex. The left column (green) represents the male sample and the right column (orange) represents the female sample. Only the female regression models were significant.

better explain these differences in both regressions. GV only accounts for 31.1% of the $R^2$ in the OAP regression and 16.1% of the $R^2$ in the outlet regression. There are no significant relationships between outlet linear dimensions or shape in males.

Sacral position also exhibits a significant relationship with GV and body mass in females, but not in males. GV and body mass account for 16% of the position of the apex of S5. GV accounts for 68.1% of $R^2$ in the regression model, meaning it has a bigger effect compared to body mass. As female gut size increases, S5 is positioned farther away from the ischial spines (Fig 5B).

## Discussion

Our study tests the pelvic floor hypothesis proposed by Abitbol [5]: in humans, we should expect to find larger gut sizes associated with narrower pelvic canals. In particular, Abitbol [5]

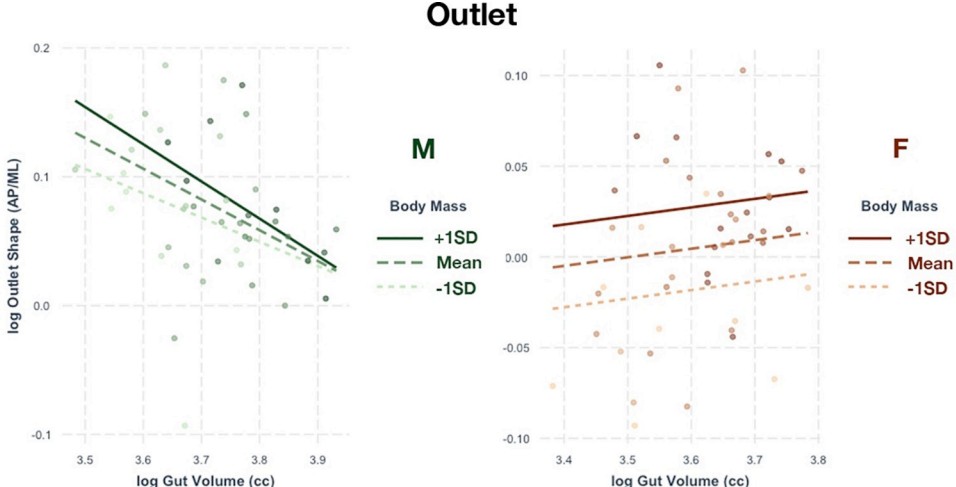

**Fig 6. Relationship of outlet shape with gut volume.** Regressions between gut volume and pelvic canal shape at the outlet relative to body mass. Only the female sample (right, orange) shows a significant relationship.

states that the abdominal weight is supported by the frequent contraction of the ischiococcygeous and iliococcygeus, which attach to the ischial spines, resulting in a narrower midplane. A narrower midplane and outlet may decrease the risk of pelvic floor disorders for those with large abdominal weights (GV). Because the occurrence of pelvic floor disorders differs between males and females [34], we tested for sex differences in visceral support mechanisms. Our results confirm differing patterns of relationship between gut size and canal dimensions in males and females. We would like to note that the $R^2$ values indicate that gut size variation explains at most 25% of male canal variation and 24% of female canal variation (Table 4). The discussion that follows below does not seek to minimize the influence of physiology, birth, and genetic drift on the pelvic canal [35–38], but rather describes the ways that gut size could add to the complexity of variation in the canal.

In females, we find that a more medial projection of the ischial spines relative to the sacrum is associated with larger gut sizes. In particular, we find that ischial spines become more medially-projecting as gut size increases in the female sample but not the male sample. We also find that the apex of S5 is positioned farther away from the ischial spines as gut size increases in females, but not in males. These observations in ischial spine and S5 positioning suggest that the AP dimensions of the posterior pelvic canal in females increase to accommodate increases in abdominal volume, but the ML dimension of the canal narrows. Such a pattern suggests that the iliococcygeus and ischiococcygeus, which are more ML oriented with origins at the ischial spines, may play important roles in maintaining pelvic floor integrity.

In males, we find that the pelvic inlet and midplane widen transversely as GV increases, especially among larger-bodied males (Figs 3 and 4). The effect seems to be diminished in smaller-bodied males. We find that this is in accordance with Tague and Lovejoy's [1] suggestion that a widening of the pelvis occurs partially for visceral support. However, the male pattern does not seem to support the hypothesis that narrower pelves result from larger abdominal weights.

Overall, our findings partially support Abitbol's [5] hypothesis regarding the morphology of the ischial spines supporting abdominal weight. Because the relationship between GV and $\theta_a$ only appears in the female sample, we suggest that this is an adaptation for a reduction of risk for PFDs in females, who are more likely to be affected due to pregnancy and childbirth-

related strain on the pelvic diaphragm [17, 18, 34, 39, 40]. The ischial spines play a supporting role for the pelvic diaphragm by projecting medially in humans, regardless of sex, in order to counterbalance the downward forces imposed by bipedality [4, 5, 41]. In females, this medial projection of the ischial spines increases the risk of obstructing fetal passage during birth. Female pelves would then need to evolve a sufficiently capacious birth canal and a sufficiently supported pelvic floor. Due to the ischial spines' importance in supporting the pelvic floor muscles (ischiococcygeus and iliococcygeus) and therefore abdominal weight, the medial projection of these spines occurs despite its narrowing of the pelvic midplane in females with larger gut sizes. In absolute terms, the female true pelvis is still more spacious than the male true pelvis on average, but the association between gut size and ischial spine projection only exists in females because they have an increased risk of PFDs. The relationship does not exist in males because they do not have an increased risk of PFDs despite having larger abdominal weights on average.

Additional weight in the abdominopelvic cavity associated with internal reproductive structures and organs was not accounted for in the present study. Compared to the male internal reproductive structures (e.g., prostate), the female reproductive structures contribute to a greater volume in the pelvic canal which may contribute to sex differences in canal morphology, specifically the inlet [12]. These reproductive structures change in volume throughout a female's lifetime in response to hormones and parity [12, 42]. A constrained age range of 18–25 in our sample may control for the variation in size (bone or viscera) related to age. The volume of the reproductive organs was not included in this study because their volumes are quite small relative to the volume of the intestines. For example, the average volume of the uteri from individuals 18–25 years old ranges from 49.8–59.8 cc [42], which represents about 1.2–1.4% of the 4192 cc average gut volume of female individuals in our sample (Table 1). We chose to use intestinal volume for this study because the intestines occupy the most space in the abdomen compared to any other organ and volume is a more relevant measurement for understanding the mechanical and spatial accommodations by the pelvis for the gut. The intestines have been theorized as particularly relevant to the evolution of the human condition and have been a focus of the scholarship surrounding torso skeletal evolution [6–8, 43].

Our findings indicate that the female pelvic canal both provides the space required for gut volume as well as the bony support required for the abdominal weight associated with that gut volume. The female midplane accommodates increasing abdominal weight by increasing the medial projection of the ischial spines as the ischiococcygeus and iliococcygeus experience more strain [5, 44], despite it narrowing the canal for fetal passage. This may also be why the pelvic canal is only transversely wide at the inlet while the midplane and outlet are more sagittally expanded. However, gut volume is not only accommodated by the pelvic floor; the true pelvis must still make spatial accommodations for larger gut sizes elsewhere in the canal and in the false pelvis [1]. We see this reflected in the increasing distance between S5 and the ischial spines associated with increasing gut volume in females. We also see this reflected in the male sample where increasing gut size is associated with a wider inlet and midplane shape. We conclude that there are differing support mechanisms in males and females: the female mechanism seems to reduce the risk for PFDs and the male mechanism focuses on spatial accommodation for the gut.

The patterns illustrated by our study adds complexity to the evolution of sex differences in pelvic morphology in humans. GV, with body weight as a covariate, accounts for 14–25% of variation in the pelvic canal dimensions measured here. While our study aligns with the findings from other researchers that indicate that the pelvis also evolved as bony support for the pelvic floor and abdominopelvic organs [3, 17, 18, 40, 44], we want to highlight that this may only be one factor that influences the evolution and variation in the pelvic canal. Obstetrics,

climate, ecology, physiology, and population history likely play important roles in shaping the pelvis [13–16, 35, 37, 38, 45–49]. The size of the birth canal, for example, seems to be conserved in female pelves from large- and small-bodied populations, indicating obstetrical demands remain important [49]. There are also differing developmental effects upon the female pelvis compared to the male pelvis related to bone remodeling that could be confounding observations [12, 15, 38]. Although evidence is still emerging, it seems that the female pelvis undergoes more substantial remodeling throughout life compared to the male pelvis [15, 38]. The pelvic canal is also subject to secular changes related to nutritional effects on body size [46, 47]. Birth canal shape may have also evolved through genetic drift [14, 37]. Gut size, therefore, is likely a minor though detectable influence on pelvic canal morphology in humans.

## Clinical and paleontological implications

The evolution of pelvic morphology is of interest to both anthropologists and medical professionals to provide a more comprehensive understanding of how visceral support plays a role in shaping the pelvic canal. Visceral support seems to influence pelvic evolution and should be considered in studies of human variation and evolution.

The World Health Organization [50] estimates that approximately 39% of adults are overweight (but see [51]: how BMI does not accurately represent body mass). This excess mass puts strain on the load bearing aspects of the pelvis and can result in an increased risk of urinary incontinence and pelvic organ prolapse [52, 53]. Due to the plastic nature of the pelvic morphology and the increasing body weights in younger children [50], clinicians should be aware of the morphological adaptations in the pelvis to decrease the risk of PFDs including decreased dimensions in the true pelvis.

Occurrence of PFDs is also influenced by the lumbosacral angle and the orientation of the pelvic inlet [44]. Women with an increase in lumbosacral angle (due to increased lumbar lordosis) and a more vertically oriented inlet exhibit lower occurrences of pelvic prolapse [44, 53]. Cultural, developmental, and ecological variation also influence pelvic canal morphology, as well as maternal health [2, 15, 45–48]. Therefore, it is unclear how much PFDs influence differences in fertility and mortality in ways that drive natural selection.

The findings in this study also have implications for our understanding of the evolution of the human pelvis and the way hominin fossil pelves are interpreted. Tague and Lovejoy [1] have suggested that the shape of the hominin bipedal pelvis, particularly the platypelloid pelvis of *Australopithecus*, evolved partially to accommodate large abdominal viscera. As hominin gut volume hypothetically decreased and brain size increased [7], the pelvic morphology also changed to a more sagittally-expanded pelvic canal [54]. Whether this change is due to changes in gut volume, fetal size, or both is unclear. If the hypothesis that bipedal hominins like early *Australopithecus* had large gut sizes and somewhat large neonates [7, 55], then they would have experienced similar pressures [56]. The fossil pelvis from *Australopithecus afarensis* A.L. 288–1 is "funnel shaped" and has ischial spines that project medially, similar to *Homo sapiens* [56]. The assumption that certain pelvic types occur with certain gut sizes (e.g., "broad" pelves occur with large guts) happens with some frequency in the paleoanthropological literature [6–8]. However, based on the small amount of canal variation explained by gut size, it seems that gut size cannot be reliably predicted using pelvic types, especially considering that sexual dimorphism is also a confounding factor on the gut-pelvis relationship [9]. Further, we do not know the extent to which the findings in this study can be used to infer gut-pelvic relationships in other species of hominins. A more comprehensive understanding of the evolution of pelvic morphology and gut size variation can aid in our understanding how and why we see these different pelvic morphologies developing in relation to gut volume in fossil hominins and modern humans.

## Limitations

Our dataset lacks some background information about the individuals in the archival CT scans amassed for the study. Information about the subjects' diet and prior pregnancies was not available. Our selected narrow age range of 18–25 may decrease the chance of having included multiparous subjects. Changes in the microstructure of the gut related to consumption of high fiber diets have been documented [57–59], but changes in the gross anatomy of the gut (such as changes in its length and weight) related to diet have not been reported in the literature. We were also unable to account for climatic variables and population history due to the lack of information about geographic origin or ancestry. Furthermore, as Betti [45] highlights, most samples used in studies on the pelvic canal overrepresent people of European ancestry since most studies are conducted within W.E.I.R.D. (Western, Educated, Industrialized, Rich, and Democratic) countries [60]. Our study is likely no exception because our sample was from Madison, WI, USA. The conclusions presented here cannot inform us about how the canal varies relative to gut size in groups of differing ancestry or geography [2]. These unknowns could be obscuring some patterns in our data.

## Acknowledgments

We thank the University of Wisconsin School of Medicine and Public Health and John Hawks for enabling access to data used in this study. We presented this research at the American Association of Physical Anthropologists 2021 Meeting and we appreciate the constructive conversations during and after the session. We thank Kirsten Brown and two anonymous reviewers for their helpful critiques that improved this paper.

## Author Contributions

**Conceptualization:** Jeanelle Uy, Natalie M. Laudicina.

**Formal analysis:** Jeanelle Uy.

**Investigation:** Jeanelle Uy, Natalie M. Laudicina.

**Methodology:** Jeanelle Uy, Natalie M. Laudicina.

**Visualization:** Jeanelle Uy.

**Writing – original draft:** Jeanelle Uy, Natalie M. Laudicina.

**Writing – review & editing:** Jeanelle Uy, Natalie M. Laudicina.

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
