## [Decision Letter · Decision Letter 0]

30 Mar 2021

PONE-D-21-04135

Assessing the role of the pelvic canal in supporting the gut in humans

PLOS ONE

Dear Dr. Uy,

Thank you for submitting your manuscript to PLOS ONE. After careful consideration, we feel that it has merit but does not fully meet PLOS ONE’s publication criteria as it currently stands. Therefore, we invite you to submit a revised version of the manuscript that addresses the points raised during the review process.

We look forward to receiving your revised manuscript.

Kind regards,

Karen Rosenberg

Academic Editor

PLOS ONE

Journal Requirements:

Additional Editor Comments (if provided):

We have received three thorough and helpful constructive reviews of your manuscript. All reviewers are generally positive but raise a number of issues, mostly methodological. Please address these issues, either by making the changes the reviewers suggest, or explaining n a letter to us why you have chosen not to do that.

Reviewers' comments:

Reviewer's Responses to Questions

**Comments to the Author**

1. Is the manuscript technically sound, and do the data support the conclusions?

Reviewer #1: Partly

Reviewer #2: Yes

Reviewer #3: Yes

2. Has the statistical analysis been performed appropriately and rigorously? 

Reviewer #1: No

Reviewer #2: Yes

Reviewer #3: Yes

3. Have the authors made all data underlying the findings in their manuscript fully available?

Reviewer #1: Yes

Reviewer #2: Yes

Reviewer #3: Yes

4. Is the manuscript presented in an intelligible fashion and written in standard English?

Reviewer #1: Yes

Reviewer #2: Yes

Reviewer #3: Yes

5. Review Comments to the Author

Reviewer #1: The manuscript by Uy and Laudicina aims to test the “pelvic floor hypothesis” by measuring the relationships between gut volume, as a key factor in the weight acting on the pelvic floor, and measurements of pelvic canal size and shape in females and males. Overall, the approach is a logical test of the hypothesis, however, the study suffers from potential statistical design issues, over-interpretation of the results, and a lack of clarity on some methodological decisions. I have organized my comments into “major” and “minor” points.

Major Comments

1. The main issue is that the statistical results do not support a robust causal interpretation of the relationship between gut volume and pelvic canal measures. That is, the relatively weak relationships detected are not strong support for a causal role for gut volume in determining pelvic size or shape, including ischial spine projection. One key issue is that there are 11 relationships tests for each sex, which is a situation where p-values should be adjusted for multiple comparisons. That is, the risk of making a type I statistical error is increased when multiple tests are done. An alpha value of p < 0.05 is arguably too relaxed in this situation. A more conservative p< 0.01 could be adopted, or p-values could be adjusted for multiple comparisons using Holm’s adjustment or some other. Assuming the authors are using R for their analysis (since they use geomorph for their landmarks), such an adjustment can be made using the function p.adjust. Likely, most of the significant relationships will become non-significant with the adjusted p-values.

But this also relies too much on statistical significance as a threshold. The more informative coefficient in this context is r-squared, since it is an effect size measure. R-squared for the regressions that the authors identify as significant range from 0.10 to 0.16, meaning these pelvic dimensions/shape account for only 10-16% of the variation in gut volume. That is not very much, and again suggests that the role of gut size in influencing pelvic capacity is minimal.

2. A related point with respect to the statistical design is how the regressions were carried out. It appears from the way the results are presented and discussed that gut volume was regressed on pelvic dimensions. This is effectively testing whether pelvic dimensions explain variation in gut volume, which seems to flip the intended causal interpretation. From the discussion of the pelvic floor hypothesis, it seems that the point is that gut volume explains pelvic floor size variation, not the other way around. Perhaps I am just interpreting the described methods incorrectly, but if so, then more clarity is needed in outlining the methods.

3. I understand the gut volume is the major component of soft tissue in the abdomen and therefore arguably could play a large role in weight on the pelvic floor. However, what are the implications of ignoring differences between females and males in the other soft tissue structures. While structures like the bladder are likely similar between the sexes, the reproductive organs are not. The uterus in particular is a relatively large structure found only in one sex (see next comment).

4. Pubic Symphysis Width: Why is this measure included? It appears to be a measure of joint breadth only, rather than capturing some meaningful aspect of pelvic capacity. Or, if it does, please explain. What is it supposed to reflect? This is particularly important since it is a significant variable in females in the results (but see comments above). The authors interpret (page 12) that this finding relates to increased gut size, but how much does this measure actually increase the volume of the pelvic canal?

5. The angle reflecting medial projection of the ischial spines: While there is a logic to the notion that this angle relates to breadth of the midplane (distance between ischial spines), what would be the geometric effects of a change in position of S5 apex on the angle? That is, holding inter-spinal distance constant, but if the apex moved posteriorly (or anteriorly) resulting in an AP expanded (or reduced) midplane. Why does this angle measurement reflect pelvic floor size better than an inter-spinous dimension, or coupled with a posterior midplane dimension (between landmark 1-7, and 2-7)? The point being that as several authors have shown (e.g., Tague, Brown, etc.), anterior and posterior spaces of the pelvic canal are sexually dimorphic and therefore may reflect pelvic floor differences that may be important here. How does this angle capture that?

In light these points, the authors should be more cautious about their overall interpretations in the Discussion section. I’m not sure they can definitively argue that they have found support for the pelvic floor hypothesis, at least not strong support. The relationship between gut volume and pelvic size/shape may just as easily be due to allometric effects – larger bodies have larger guts and larger pelves. The authors should also consider the model recently presented by Dunsworth (2020. Evolutionary Anthropology, 29:108-116). In this article, Dunsworth presents a related argument about the importance of considering the soft tissues of the pelvis and their role in influencing pelvic growth. However, Dunsworth relates the influence to the effects of estrogen more specifically. She also provides useful data on the size of other organs, such as the uterus, which relate to my point 3 above.

Minor Comments

Introduction. The anatomy of the pelvis (true and false pelvis) is briefly discussed. A diagram illustrating these regions would be helpful for those readers without a background knowledge of pelvic anatomy, particularly of the 3 planes that are included in the study.

Line 60: The 3 pelvic planes are described as “the most constricted locations”, but they are not, they are just clinically identified planes of the canal marking the entrance, middle (which is most constricted), and exit of the canal.

Line 65: “morphology pelvis” should be “morphology of the pelvis”

Lines 76-78: “These pelvis components develop early”, perhaps should be “begin to develop” since development is not completed by birth? Also “ossification occurs every month until birth” is confusing. It implies that it occurs once a month rather than continually. Might be better described as “continuing throughout childhood and into adolescence”? The authors might also not that there are a number of secondary ossification centers and the timing of fusing of all these components is complex and not completed until the 3rd decade of life.

Line 91: Do Tague and Lovejoy really say the “main role” of the pelvis is visceral support? Arguably the main role of the pelvis is locomotion?

Line 101: What does “no observed abnormalities” mean? What would have constituted and abnormality? Would a shape abnormality, such as a woman with abnormally narrow pelvis have been excluded? This is particularly relevant to whether the sample is capturing “normal” variation or excluding some.

Line 160: “across sexes” should be “between sexes”.

Results: A table of descriptive statistics for the sample variables would be a useful addition.

Throughout the Methods and Results sections, the authors repeatedly refer to the “correlations” between gut volume and pelvic measures. However, the analysis conducted was regression. While correlation and regression are related methods, and both look for associations between variables, they are not the same. Correlation has a particular meaning, statistically, so it would be better to say “relationship” or “association” unless correlation analysis, specifically, was used.

It is not clear why Table S1 and Figures S1 and S2 are not included in the main body of the manuscript. The table in particular would be valuable since it provides important definitions of the landmarks.

Line 233: “adds” should be “add”

Reviewer #2: General Summary

Dr. Jeanelle Uy and Natalie M. Laudicina

Dr. Uy and Ms. Laudicina examine the relationship between gut volumen and different measures of the true pelvis (AP and ML dimensions). Based on papers from Abitbol (1988) and Tague and Lovely (1986), they hypothesized that gut volume will show significant negative correlations with dimensions associated with the pelvic floor in females but not in males. The justification for this hypothesis is that in females the pelvic dimensions are minimized to support viscera and facilitate birth, which is not the case in males. In order to test this hypothesis, they extracted 3D landmarks, their associated interlandmark linear distances, an angle (Θ) from distances, and gut volume from a clinical CT sample of male and females. The authors then performed linear regressions between gut volume and pelvic dimensions in both male and female samples. Results demonstrated that in females larger gut sizes were correlated with more medially projected spines, but not a narrow ML midplane. In contrast, larger gut sizes were correlated with wider pelvic dimensions in males. Results partially partially supported their original hypothesis. In conclusion, the authors argue that negative correlation between Θ and gut size indicates an adaptation for visceral floor support and to minimize pelvic floor disorders. Overall the manuscript is publishable, although I recommend several areas and additions that should be considered.

Major Issues

Methods

1. The clinical sample is poorly defined, with the exception of an age range and sex. If factors such as prior pregnancies, self-reported race, height, and weight are available, then these need to be presented, perhaps in the form of a table (with means and standard deviations). Otherwise this limitation needs to be acknowledged in a limitations section. Based on a related paper from the same sample, some of this data may be available and some may not.

2. There is no mention of an error analysis or similar study for the landmarks and the volumetric data. There can be a high degree of variability in isolating landmarks, particularly for those areas that are more difficult to define. These areas are usually sampled several times and rendered as “fuzzy” (Type III) landmarks (Bookstein, 1991). I see no mention of what if any pelvic landmarks would fall into that. What may seem like minor differences in how a given landmark is approximated, can result in much larger differences when 3D shape and/or interlandmark linear distances are considered. Moreover, these distances are more pronounced for smaller distances. This is critical especially if more than one of the authors segmented the CT data, performed the landmarking, and segmented the volumetric data. Intraobserver and interobserver reliability and precision should be referenced (Kohn and Cheverud, 1992)

Results

1. I’m unclear to why body size, if available, was not at least included as a covariate in the analysis. If it wasn’t available, then it needs to be included as one of the limitations. If it is available, then it should be included as a confounding variable. The remaining question I have is would the analyses have changed if body sized proxies were included? This seems like a huge gap, especially considering the influence of body size on pelvic dimensions and size. If the authors run the analyses and find that there is no effect, then body size can be eliminated as a factor; however this still needs to be addressed and clarified for the readers.

2. Significant correlations were identified between several dimensions and gut volume in males and females. While the R2 were included, I would have preferred to see the authors couch the significance in terms of the overall effect. The highest R2 for any of the significant analyses was 0.16, demonstrating that there is still an enormous amount of variation not accounted for solely by pelvic dimensions. That’s not surprising per se given the literature, but each of the significant findings should refer to the strength of these associations in terms of biological meaningfulness (e.g., significant small positive or significant moderate negative effect). As the authors emphasize later, there are numerous theories and factors that have been theorized to be related to gut size. The relationship between gut size and pelvic dimensions is significant but not likely the sole explanation and the reason there is some spread in the overall data. This would help the authors contextualize the findings to larger issues of gut and pelvic evolution given the multitude of factors involved.

3. One of the challenges in looking at distance-based metrics for 3D structures is that the context between related points are lost. What was a complex plane through the bony pelvis, is transformed into a ML and an AP dimension. Examining the linear distances is appropriate for the analyses; however there is complex shape information that can be lost. For example, the Θ was calculated based on the angle between the right and left linear distances from the ischial spines to the apex of S5. A wider angle would suggest that the spines are more laterally displaced; however the angle is formed from a 3D structure, with the spines and the apex of S5. My question is what the actual shape looks like here - is there concurrent displacement of the apex of S5 with the movement of the ischial spines? How does that dimension related to the angle calculated? In other words if you posteriorly displace the sacrum but keep the ischial spines in the same points, do you get different angles. In a 3D space, the difference might be attributed to the movement of the sacrum and the spines. The authors findings that ischial spine projection was negatively correlated with gut size, but the MML was not lend potential support to a more complex shape configuration. It might be worth looking to see whether just the posterior outlet (spine-S4S5) shows any differences in correlation with gut size. This gets to a larger discussion of skeletal adaptation within the pelvis: how to accommodate the myriad selective pressures on pelvic landmarks given 3D constraints. How is the shape of the pelvis changing to result in wider or narrow dimensions, and is there more than one way to accomplish that?

Discussion

1. Per my previous suggestion of the relative effect size, the results should be emphasized in relation to biological significance. The conclusions regarding significance and relationship between the data are supported, but the degree to which these factors affect are not presented. This will also provide a richer discussion for examining this complex relationship.

2. Overall, I think the discussion can be expanded upon. Per my previous comment there’s an incredible degree of

variation in gut size that is not explained by the relationship between the pelvic dimensions. At the heart of both pelvic shape and gut volume are incredibly complex interplay between many different, sometimes competing, selective pressures. That’s what makes this type of research, in my opinion, so interesting. The authors briefly mention some theories, but expanding on the theories would make the paper stronger.

3. There needs to be a limitation section. First, as detailed there are questions about the details of the clinical sample. Second, these results may or may not hold up when looking at other samples (e.g., those of very different body proportions and/or sizes), so the generalizability is limited without broader samples. However, the transferability of the research in reference to the fossil record is a nice point for future directions.

Reviewer #3: “Assessing the role of the pelvic canal in supporting the gut in humans” by Drs. Jeanelle Uy and Natalie Laudicina is a fine manuscript that warrants publication. This paper will be of interest to a number of researchers, and this paper will be cited frequently in the literature. The hypothesis is clearly stated with sufficient background explanation; the materials and methods are clear; the results are succinctly presented; and the discussion is straightforward and in line with results. I have one principal recommended revision and a few minor editorial suggestions.

Principal comment: As a review of the manuscript, I am not “requiring” that the authors follow my recommendation, but I believe that doing so clarifies the biology that they present. The authors present 22 results of regression analysis in Table 2. The authors report two statistical results as being nonsignificant with the following p values: (1) female, IML with p=0.05, and (2) female, Outlet (ap/ml) with p=0.051. My comment is that if the authors are going to be strictly binary with p<0.05 as significant (and the authors should state in Materials and Methods the alpha level that they choose), then perhaps the authors should also adjust the level of critical significance by using a methodology to reduce the likelihood of Type I error. The authors perform 22 statistical tests and employ p<0.05 for each test, but that multitude of tests increases the chance of Type I error. Bonferroni’s method is one way to adjust the critical alpha level for multiple tests. Using Bonferroni’s correction, the critical alpha level for significance becomes 0.002 (i.e., 0.05/22). If Bonferroni’s correction is used, then none of the statistical tests in this study is statistically significant, and that obviously meaningfully changes the authors’ interpretations. I am not a supporter of Bonferroni’s method because it yields a critical alpha level that is difficult to satisfy when sample size is small, which is often the case in anthropology. I see two alternatives to this muddle. First, the authors can state affirmatively in the text that the results for female IML and female Outlet (ap/ml) closely approach significance, and include these two variables in the discussion of results. That is, the authors can report that results “provisionally” suggest that IML and Outlet (ap/ml) are positively related to gut volume in both females and males. The second alternative is for the authors to state in M&M that they are using an alpha level of ≤0.05 rather than the implied <0.05. With an alpha level of ≤0.05, the probability levels for both female IML and female Outlet (ap/ml) are significant. Female IML is significant because its p=0.05; female Outlet (ap/ml) would be significant because its p=0.051 rounds to 0.05. To be consistent in reporting the probability levels for the 22 results in Table 2 with respect to the alpha level of ≤0.05, then either actually or implicitly round all probabilities to the 2nd decimal place. This approach is not deceptive or manipulative; rather, it is fully appropriate. When a statistician writes that the alpha level is 0.05, the inference is that p=0.05 is significant (i.e., a statement that the alpha level is set to have the likelihood of Type I error being 5% means 1/20 or 0.05, not only values less than 0.05). Correspondingly, if one reports data to the 2nd decimal place, then all values from 0.495 to 0.054 round to 0.05 and, therefore p=0.051 is actually 0.05. Using either of these alternatives does not meaningfully alter the authors’ interpretations. Rather, using either alternative shows that females and males are not meaningfully different in the relationship of IML and Outlet (ap/ml) with gut volume, and results of this study really support this interpretation. Indeed, if I were to cite this study in one of my papers, I likely would qualify that citation by stating that results of this study really show that the sexes are comparable in the relationship of gut volume with these two variables that I mention, despite the authors’ decision not to recognize that relationship.

More than a minor comment but less than a principal comment: The authors probably should cite the following paper in their manuscript: Baragi, DeLancey, Caspari, Howard and Ashton-Miller. 2002. Differences in pelvic floor area between African American and European American Women. American Journal of Obstetrics and Gynecology 187(1):111-115.

Minor suggested changes:

line 64: Change “intestines (“gut volume” or “gut size”) therefore its size” to “intestines (“gut volume” or “gut size”); therefore, its size.”

line 65: Insert “of the” between “morphology pelvis” – “morphology of the pelvis.”

line 102: Replace semicolon with comma: change “individuals aged 18-25; as a narrow” to “individuals aged 18-25, as a narrow.”

line 271: Insert comma after “Australopithecus.”

Again, congratulations to the authors for a fine study and well written manuscript.

6. PLOS authors have the option to publish the peer review history of their article (what does this mean?). If published, this will include your full peer review and any attached files.

Reviewer #1: No

Reviewer #2: **Yes: **Kirsten Brown

Reviewer #3: No

---

## [Author Response · Author response to Decision Letter 0]

12 May 2021

Thank you for the opportunity to revise this manuscript and for the constructive reviews. We have given a detailed account of our revisions in the "Response to Reviewers" document. We think that the changes, though major, did not substantially change our original conclusions—the results still reflect differing support mechanisms in males and females. The female mechanism reduces their risk for pelvic floor disorders and the male mechanism focusing on spatial accommodation for the gut.

---

## [Decision Letter · Decision Letter 1]

5 Jul 2021

PONE-D-21-04135R1

Assessing the role of the pelvic canal in supporting the gut in humans

PLOS ONE

Dear Dr. Uy,

Thank you for submitting your manuscript to PLOS ONE. After careful consideration, we feel that it has merit but does not fully meet PLOS ONE’s publication criteria as it currently stands. Therefore, we invite you to submit a revised version of the manuscript that addresses the points raised during the review process.

We look forward to receiving your revised manuscript.

Kind regards,

Karen Rosenberg

Academic Editor

PLOS ONE

Journal Requirements:

Additional Editor Comments (if provided):

All reviewers see that the manuscript is significantly improved and all offer minor suggestions for revision. I would add a couple of small points not covered by the three thorough and helpful reviews.

Line 45 -- the authors describe the dual roles of the pelvic canal as parturition and visceral support but isn't it also possible that locomotor constraints are an influence, even of the canal by constraining other aspects of pelvic morphology?

Line 181 "a technique commonly used in radiology and [that] has been described..."

Line 192 "as" should be "has"

Line 378: I don't understand this sentence "The birth canal also shows a signal of neutral evolution, climate-related variation and natural selection when considering a global sample."

Line 490 There seems to be something wrong with the first authors name in this reference. I think probably delete the "9"

I found some of the lettering on the figures too small to see.

Reviewers' comments:

Reviewer's Responses to Questions

**Comments to the Author**

1. If the authors have adequately addressed your comments raised in a previous round of review and you feel that this manuscript is now acceptable for publication, you may indicate that here to bypass the “Comments to the Author” section, enter your conflict of interest statement in the “Confidential to Editor” section, and submit your "Accept" recommendation.

Reviewer #1: (No Response)

Reviewer #2: All comments have been addressed

Reviewer #3: (No Response)

2. Is the manuscript technically sound, and do the data support the conclusions?

Reviewer #1: Yes

Reviewer #2: Yes

Reviewer #3: Yes

3. Has the statistical analysis been performed appropriately and rigorously? 

Reviewer #1: Yes

Reviewer #2: Yes

Reviewer #3: Yes

4. Have the authors made all data underlying the findings in their manuscript fully available?

Reviewer #1: Yes

Reviewer #2: Yes

Reviewer #3: Yes

5. Is the manuscript presented in an intelligible fashion and written in standard English?

Reviewer #1: Yes

Reviewer #2: Yes

Reviewer #3: Yes

6. Review Comments to the Author

Reviewer #1: Drs. Uy and Laudicina have done an excellent job revising this manuscript. The statistical approach is more logical now, and the interpretation of the results more tempered. Overall, the research is more clearly presented and discussed, and I thank the authors for taking the time to complete these revisions. My comments below mainly relate to a few aspects of how the results are presented – certain coefficients that probably should be provided are not, and more clarity on what Figures 3 and 4 are showing.

1. The multiple regression analyses are useful as they consider the influence of both body size (mass) and gut volume on the pelvic dimensions. As well, the more conservative alpha (0.01) is a good idea, but I also dislike a strong adherence to p-values as arbiters of “importance” on their own. The R-square value is just as important as an indicator of the relationship. With this in mind, and looking at the patterns of p-values and R-square values in Table 4, I think the authors could note that MAP (in females) and MML (in males) also show a relationship as a less conservative alpha (0.05), through with lower R-square values. Not to dwell too much on this, but it suggests there may be something happening differently in females and males for the midplane (which dimension is influenced by GV), since both sexes show a relationship with midplane shape.

2. I am, however, a bit confused by the results presented in Table 4. The slope for each predictor (body mass and GV) would be more useful coefficients to report than the intercept for the model – the slope tells us which direction the relationship between predictor and predicted variables is. The intercept doesn’t really tell us much of interest. The p-value reported, is that for the F-test from the regression or for the intercept? If the latter, this p-value does not tell us about the relationship, it only tests whether the intercept = 0. If it is the F-Test p-value, then it just tells us the regression model is predictive, but not about the null hypotheses concerning the individual predictor variables in the model. It would be more useful to have the slopes and p-values for each predictor. I recognize that the relative importance analysis (Table 5) tells us about which predictor is more important, which is useful, but the model coefficients for the predictors are also useful information.

3. Page 15, line 226: “it seems the inlet becomes mediolaterally wider as gut size increases” – should this be “relatively” mediolaterally wider? A decrease in the ration of AP/ML can be brought on by an increase in ML or a decrease in AP; that is, ML could stay the same in absolute length, but relative to AP it increases. The same would be true for the other shape relationships discussed. Except perhaps midplane – as per my comments above about the hint of a relationship of MML and MAP in males and females, respectively. Here, whether ML or AP changes may be more interpretable for each sex based.

4. Figures 3 and 4 – I don’t quite understand the fit lines in these graphs. How is mean, -1SD and +1SD body mass being plotted each as a line through the log-shape vs. log-GV scatter? Each of these (mean, -1SD, +1SD) is a single value of body mass, is it not? And what then are the fit lines representing with respect to the relationship between shape and GV? Perhaps just a clearer caption explaining the elements in these graphs would help.

5. Page 18, lines 290-295: I like this statement, it recognizes the complexity of factors shaping the pelvis without diminishing the results of this study. And the authors nicely contextualize these other factors in the next section of the Discussion, so thanks!

Reviewer #2: General Summary

Dr. Jeanelle Uy and Natalie M. Laudicina,

Dr. Uy and Ms. Laudicina have made significant changes in their original manuscript. They have adequately addressed reviewers' concerns and suggestions to the best of their ability and within the scope of the manuscript. Entire sections (e.g., results ) were rewritten and other sections were changed substantially. Importantly and at the behest of several reviewers, they hedged their conclusions and discussion to address the lack of clear evidence for some of these complex relationships between gut, pelvic, and body size. The modifications have improved the readability and made the manuscript more appropriate for a general audience, as opposed to those of us who focus on pelvic morphology. I provide a minor, but not required, suggestion for publication. The rest of my commentary is for their review only. I look forward to the publication in the future!

Minor Changes recommended:

Discussion, Line 288: Citation would be useful for “occurrence of pelvic floor disorders differ between males and females….” I know this is true, but is there a paper to cite?

Comments regarding each section

Abstract

The rephrasing of a relationship vs. a correlation between was a welcome change. I realize the authors addressed this as a change in results, but it makes the case clear from the onset. From a semantic standpoint there is a difference in how readers perceive the two words. Given the ambiguity in so much of this topic, hedging is the right call.

All other changes look good.

Introduction

Lines 70-72: Kudos to the authors (and R1 for the suggestion) for adding the figure of the three pelvic planes! It did not crossed my mind but is appropriate given PLOSOne audience.

Lines 104-105: Clarification of size versus shape was a major sticking point of mine in the original review. So many times, these two terms, especially in the pelvis, are equated as the same. While related due to allometry, etc., these are not the same. Authors need to read very carefully when using these terms. This is further complicated when comparing linear measurements and body size proxies. I appreciate the authors’ addition in parsing these terms out.

Methods

Line 130: Description of the clinical sample, including addressing the abnormalities (R1) is acceptable. As is the case with clinical samples, the background information is limited. Addressing this here but also in the limitations is acceptable.

Lines 148-158: Error analysis, fuzzy landmarks, and the averaging of distances from landmarks address my concerns regarding error in 3D landmarks and corresponding error in calculated distances.

Line 144: I was pleased to see R1 also raised concerns regarding the challenge in quantifying shape and positions of the ischial spines based on a single angle. Without looking at 3D morphospace of the pelvis, the addition of the Θa was an improvement on most linear based studies that only look at maximum dimension (e.g., MML). As the authors noted, the angle was an attempt to quantify more of that 3D shape. The original critique I had was there was no way of knowing how the rest of the sacrum was positioned relative to the ischial spines. Adding the Θb at the level of S5 more adequately captures some of the pelvic shape and provides a more direct measure of pelvic floor shape. This was a nice addition when 3D morphospace examination was not feasible.

Line 188-205: The new statistical analyses section is much improved. Regressing the canal measurements on gut volume provides a clearer interpretation of the pelvic floor hypothesis and makes the subsequent sections more readable. I am pleased to see the authors included body size as a covariate (interacting for males based on previous literature vs. non-interacting for females), addressing the issues I raised in my first review. Pelvic size (perhaps shape) and body size are related in some ways, but not necessarily others depending upon the measures and sexes you examine. However, the very least that needs to be done is to examine whether there is an effect before deciding not to include it at all. Similarly the post-hoc analysis looking at the relative weights on the observed variation in R2 among dimensions was a nice compromise to this complex issue.

Results

Overall dividing the results into the major pelvic planes drastically improves the readability. There’s a lot to parse out and having these organized similarly to how the charts are presented makes it more approachable, particularly for a general audience.

As I mentioned in the above, the relative importance of GV and body weight on R2 significantly adds to the paper. It also helps to contextualize the strength of the associations. Overall this is small to moderate effects (not surprising given all of the forces acting on the pelvis);however the differences in the relative contributions is stark. Kudos to the authors for the addition.

Discussion

Hedging of the relative influence of gut size on pelvic variation is a nice addition.

What really comes out with the added analyses and discussion is the ability to more adequately visualize and quantify the 3D morphospace. In reading the male and female differences in the lower pelvic planes, the “movement” of the ischial spines relative to the sacrum, which elongates the pelvic floor, is much clearer in this revision than in the first draft. That 3D shape change is reflected in cited works in the manuscript as well.

Line 362-380: A succinct summary of the myriad of factors that act on the pelvis. Well done! For a general audience, this also provides a starting point for others to delve into these issues more, should they be inclined. It really does tie the picture together and adds to our understanding of this system.

Lines 422-438: Limitations section adequately addressed my comment in the original review.

Reviewer #3: Recommendation: Accept with very minor revisions.

Perhaps in accommodating reviewers’ recommendations of the original manuscript, the authors diminish the results and interpretations of their study: “Gut size . . . is likely a minor though detectable influence on pelvic canal morphology in humans” (lines 379-380). The authors further minimize their study by listing multiple variables that influence pelvic morphology (e.g., “cultural, developmental, and ecological variation . . ., as well as maternal health”; lines 396-397; and these variables do influence pelvic morphology) and by concluding with a paragraph of “Limitations.” I do not ask the authors to make any change here, but the reader is left with a rather vacant feeling about the important results of the study as the authors end their paper by “back tracking” on the importance of their study. Undoubtedly multiple variables influence ischial spine projection and S5 positon, but this study shows that gut volume “explains” 8% and 11% of variation in ischial spine projection and S5 position, respectively (8% and 11% computed by multiplying R2 and % of R2 accounted for by GV, Tables 4 and 5). I doubt that quantifiable “cultural, developmental, and ecological variation . . . (and) maternal health” would show any higher percentage of variation explained than that of gut volume. Given the crudeness of the data that we often use, these percentages of variation “explained” are not inconsequential. I believe that the authors could have been more assertive about the results and interpretations of their study. However, I ask for the authors to make no change with respect to these comments. This is a fine study and deserving of publication. This paper should be cited in future publications of pelvic morphology.

I accept the authors’ statement that two individuals can have identical midplane ML yet different medial projection of ischial spines, but I do not understand how that difference is detectable based on the description of the variables and Figure 2. Are not midplane ML and medial projection of ischial spines both taken from the most medial point of the ischial spines (“Dorsal-most projection of the [ischial] spine located of the posterior ischium”; Table 2)? To determine medial projection of the ischial spines, it seems as if the ML measurement should be to the base of the ischial spines and the medial projection to the medial point of the spines. I also cannot discern from the angle shown in Figure 2 how the anterior projection of S5 is determined. The distance between S5 apex and ischial spine is determined by the combination of S5 anterior projection and ML length. How was ML length controlled to calculate S5 anterior projection?

I have a few minor editorial suggestions (but distracting that two authors could not see seven subject-verb disagreements in their writing).

Line 71: Either insert semicolon and comma to precede and succeed, respectively, “therefore”; or, insert end of sentence period before “therefore” and begin new sentence.

Lines 81-82: Not clear to what “effects” the authors are referring that “the pelvis . . . adapt(s) to a variety of effects.”

Line 98: Substitute “is” for “are”: “morphology . . . is” (“ischial spines” in parentheses is not the subject of the sentence).

Lines 98-99: Delete either “if” or “how”: we evaluate if how.”

Line 116: Probably insert space between comma and”20-21” in the citations, “[15,20-21].”

Table 3: In the description of S5 position, change “spine” to “spines.” Also, the descriptions for the medial projection of ischial spines and S5 position are identical. Should not the descriptions be different because these are different angles?

Lines 157-158: Disagreement in number between noun (“Gvs” – plural) and pronoun (“it” – singular).

Table 5: Perhaps round the % of R2 account for by body weight for pelvic midplane to 22.8% (i.e., to one decimal place) to be consistent with other values in the table.

Line 288: subject-verb disagreement, “occurrence . . . differ”; change to “differs.”

Lines 292 and 294: subject-verb disagreement, “discussion . . . describe”; change to “describes.”

Lines 301-302: subject-verb disagreement, “dimensions . . . increases”; change to “increase.”

Line 337: subject-verb disagreement, “volume . . . were”; change to “was.”

Lines 344-346: subject-verb disagreement, “intestines . . . has been a focus”; change to “have been.”

Lines 384-385: Should “on” be deleted? “Visceral support seems to influence on pelvic evolution.”

Line 393: subject-verb disagreement, “Occurrence . . . are”; change to “is.”

Line 403: insert comma after “Australopithecus.”

Line 408: insert space after comma in list of citations, “[7,55].”

Lines 424-425: subject-verb disagreement, “Information . . . were”; change to “was.”

7. PLOS authors have the option to publish the peer review history of their article (what does this mean?). If published, this will include your full peer review and any attached files.

Reviewer #1: No

Reviewer #2: **Yes: **Kirsten Brown

Reviewer #3: No

---

## [Author Response · Author response to Decision Letter 1]

9 Aug 2021

We very much appreciated the encouraging reviews. We have made the suggested edits. Thank you for being thorough. The following details what has been changed in the manuscript for each section and the parentheses at the end of each bullet point indicate which reviewer(s) suggested the change. If there is no parenthetical note, then this change was made under our discretion. This list is also found in the Response to Reviewers document.

Abstract

• Line 23: Minor change in syntax for one sentence.

Introduction

• Line 45: We added a sentence acknowledging that locomotion may have also influenced the pelvic canal (Editor)

• Line 91: We changed the phrase “…the pelvis to adapt to a variety of effects” into something less vague (“factors such as birth, locomotion, and climate”) for clarity (R3)

Materials and Methods

• Line 161 Table 3 – we corrected the description for S5 Position (R3)

• Line 196: We provided a clearer explanation of how S5 projection was calculated (R3)

Results

• Line 240 Table 4: We modified this table so that the slopes for each predictor variable were reported along with their p-values. We removed the column reporting the intercept. (R1)

• We revised how the regression results were presented on Table 4 so that it appeared cleaner.

• Line 264-283: 

o We modified the Fig 3 caption to better describe the plot (R1)

o We divided up Fig 3 into 3 separate figures (Figs 3, 4, and 6) to allow for the text to appear larger (Editor)

• Line 305-306: We noted that MAP in females and MML in males show significance when using an alpha of 0.05. (R1)

Discussion

• Line 347: We added a reference to the statement that there are sex differences in the occurrence of PFDs (R2)

Other

• We added one new citation (35) in the References section. (It is cited in Discussion.)

• We modified the in-text citations as necessary throughout.

• We corrected several grammatical errors (R3).

---

## [Editor Report · Decision Letter 2]

27 Sep 2021

Assessing the role of the pelvic canal in supporting the gut in humans

PONE-D-21-04135R2

Dear Dr. Uy and coauthors,

We’re pleased to inform you that your manuscript has been judged scientifically suitable for publication and will be formally accepted for publication once it meets all outstanding technical requirements.

Kind regards,

Karen Rosenberg

Academic Editor

PLOS ONE

Additional Editor Comments (optional):

Congratulations! You've been very responsive to the reviewers comments and I think the manuscript is ready to publish.
---

## [Editor Report · Acceptance letter]

1 Oct 2021

PONE-D-21-04135R2 

Assessing the role of the pelvic canal in supporting the gut in humans 

Dear Dr. Uy:

I'm pleased to inform you that your manuscript has been deemed suitable for publication in PLOS ONE. Congratulations! Your manuscript is now with our production department. 

Kind regards, 

on behalf of

Dr. Karen Rosenberg 

Academic Editor

PLOS ONE